# How Do Consumers Perceive Cultured Meat in Croatia, Greece, and Spain?

**DOI:** 10.3390/nu13041284

**Published:** 2021-04-14

**Authors:** Paula Franceković, Lucía García-Torralba, Eleni Sakoulogeorga, Tea Vučković, Federico J. A. Perez-Cueto

**Affiliations:** Future Consumer Lab, Department of Food Science, University of Copenhagen, Rolighedsvej 26, 1958 Frederiksberg, Denmark; hqt408@alumni.ku.dk (P.F.); nzt988@alumni.ku.dk (L.G.-T.); gdr923@alumni.ku.dk (E.S.); nzw448@alumni.ku.dk (T.V.)

**Keywords:** cultured meat, meat consumption, European consumers, meat substitutes, consumer perception

## Abstract

The meat production industry is one of the leading contributors of greenhouse gas emissions. Cultured meat presents itself as a potential eco- and animal-friendly meat substitute which has the potential to eradicate animal cruelty and reduce both the environmental footprint and the risk of zoonotic illnesses, while delivering a nutrient-dense product. The purpose of this study was to investigate how consumers perceive cultured meat and if the frequency of meat consumption is related to their intention of trying or purchasing cultured meat. Data were collected online in 2020 from Croatia, Greece, and Spain. Among the 2007 respondents, three segments were identified according to meat consumption and variety, plus an a priori identified group of “non-meat eaters”. Sixty percent perceived cultured meat as kind to animals, 57% as unnatural, 45% as healthy and environmentally-friendly, 21% as disgusting, and only 16% as tasty. Although 47% of the respondents had not heard of cultured meat before, 47% would taste it and 41% would purchase it for the same price as conventional meat. This indicates that consumers from Croatia, Greece and Spain might be likely to purchase cultured meat if sold at an affordable price.

## 1. Introduction

Cultured meat, also called in vitro, artificial, synthetic or lab-grown meat, is a novel technique of producing meat by growing animal cells taken from a live animal using a biopsy done under anesthesia. It is based on tissue engineering techniques and seems to be a promising solution for producing meat, while eradicating animal cruelty and reducing both the environmental footprint and the risk of zoonotic illnesses [1].

Nowadays, a significant part of meat production comes from intensive animal farming, which results in negative effects on both animals and the environment. Steinfeld et al. [2] estimated that two-thirds of agricultural land and one-third of all land surface is used for livestock production. The continuous destruction of the land surface has a significant impact on the biodiversity of plants and animals. This kind of meat production strongly increases the depletion of natural resources and causes air and water pollution [3,4]. In addition, up to a third of greenhouse gas emissions are connected to agricultural practices, even when fossil fuel inputs are included [5]. Factory farming poses a threat both to the physical and mental wellbeing of farmed animals. Apart from the ethical concern, this can also raise risks for human welfare, since the continuous usage of antibiotics can lead to the development of antibiotic-resistant bacteria, potentially harmful for human health [6].

Efforts have been made for consumers to turn towards more plant-based diets [7], since excessive and frequent consumption of meat, especially red and processed meat, has been linked to multiple adverse health effects in humans, namely cardiovascular diseases, type 2 diabetes, and some kinds of cancer [8]. Even though there is an increase in both the production and acceptance of plant-based alternatives, numerous meat consumers refuse to decrease or abandon their meat consumption [9]. According to the European Vegetarian Union [10], only 6% of the European population follow a non-meat eating diet. Cultured meat is therefore a potential solution for consumers who do not want to remove meat from their diet [11].

Although cultured meats are not yet available in the EU market, some companies are aiming at introducing the product in the Common Market by 2023. The market for cultured meat was valued at USD 118.8 million in 2020 and is anticipated to grow at a CAGR of 14.9% to reach USD 352.4 million by 2028. (https://www.polarismarketresearch.com/industry-analysis/cultured-meat-market, last accessed on 7 April 2021) Consumer acceptance is key in order for cultured meat to become mainstream and a successful innovation. Therefore, a growing body of evidence is being used to start to address consumer perceptions towards cultured meat in the EU and beyond [12,13,14]. From a public health nutrition perspective, it is of interest to evaluate consumer acceptance towards such products in areas that have a very well-established traditional dietary pattern such as the Mediterranean diet, where whole foods or minimally-processed foods are recommended, but where animal sourced food intake has increased dramatically in the past 50 years.

The objective of this study is to identify consumer segments on the basis of their conventional meat consumption, and further investigate their awareness and perception towards cultured meat, as well as their intentions to taste or purchase it, in Croatia, Greece and Spain, countries having the Mediterranean diet in common. This study will contribute to the growing—yet still limited—body of evidence on consumers and cultured meat, and will provide strategic knowledge for innovation and public health promotion.

## 2. Materials and Methods

### 2.1. Data Collection

Quantitative data were collected through a cross-national, web-based survey distributed in Croatia, Greece and Spain. Participants were recruited anonymously through snowball sampling, for 12 days during October 2020. The target population was adults above 18 years of age. A statement regarding the confidentiality and anonymity of the data was provided before the beginning of the survey. Participants were asked to give consent for their data to be used according to the General Data Protection Regulation (GDPR), before proceeding.

SurveyXact by Rambøll, Denmark, was used to develop the instrument, distribute it, and collect the data. The questionnaire was pre-tested by the researchers in order to assess readability, the understanding of the questions, and to measure the time it took to fill it (on average 7 min). The instrument consisted of 19 questions, which were divided into the following sections: sociodemographic characteristics, meat consumption frequency and variety, awareness, perception and intention of tasting and purchasing cultured meat. The survey was created in English and then translated to Croatian, Greek and Spanish by native speakers. Following GDPR participants were informed about the objectives of the survey, the intended use of their data that were provided anonymously and their right to withdrawal from the questionnaire at any point in time, and that they had to express their consent by clicking “next”.

A food frequency questionnaire (FFQ) provided the frequency of meat consumption. The categories for different meat types were divided into: fresh-cut poultry, fresh-cut beef, fresh-cut pork, processed meat, and other types of meat. Examples of the most common meat products of each country were provided to avoid any ambiguity. Meat consumption frequency was analyzed on a 7-point scale ranging from “never” to “more than once per day”. Moreover, we needed to stress that the frequencies were not intended to evaluate actual nutritional intake or relate it to any health outcome or to infer nutrient intake. Frequencies were used as a consumer segmenting variable.

Participants were asked what would motivate them to eat less/no meat and were given the option to choose more than one answer between animal welfare, health, and environmental reasons. Awareness about the term “cultured meat” was investigated through a simple question in which the participants could choose among the answers: “I have never heard about it”, “I have heard about it”, and “I know about it”. Following that question, a definition of the term cultured meat was provided to make sure the participants had a basic understanding of it. Afterwards, participants were given a chance to write the first thing that came to their minds when they thought about “cultured meat” in an open-ended question. 

Respondents were subsequently given the following statements: “I think cultured meat is healthy/environmentally friendly/ tasty/ kind to animals/ unnatural/ disgusting”, for which they expressed perception on a five-point Likert scale, from “strongly disagree” to “strongly agree”.

The intention of tasting cultured meat was examined on a five-point scale from “very unlikely” to “very likely”. Finally, the last three questions evaluated the participants’ actual intention to buy cultured meat. Participants were asked if they would buy cultured meat at a lower, higher, or same price as conventional meat.

### 2.2. Data Analysis 

Data from SurveyXact were extracted as an Excel document and processed using RStudio 1.3. RStudio is an integrated development environment (IDE) for R by RStudio a B Corporation in Boston, MA, USA.

The information acquired from the meat consumption FFQ was used to calculate the total frequency of meat consumption. First, “non-meat eaters” were grouped in a separate category based on their reported zero meat consumption frequency. Subsequently, meat-eaters were subjected to k-means cluster analysis using total meat consumption frequency and processed meat consumption as variables. This resulted in three clusters: “medium frequency, low processed meat”, “high frequency, medium processed meat”, and “very high frequency, high processed meat”, plus the a priori identified group of “non-meat eaters”. 

Socio-demographic characteristics are presented as means (SDs) or as percentages. A chi-square test was used to assess differences between categorical variables, while ANOVA was used for continuous variables (e.g., age). Multivariate logistic regression analysis was performed to identify the attitudinal and socio-demographic characteristics associated to cluster belonging. For that purpose, dummy variables (1 = belong to the cluster, 0 = does not belong to the cluster) were created to indicate cluster belonging. Binary logistic regression models were fitted with each dummy variable as dependent, and as explanatory variables: First, the sociodemographic data age (1 y increment), sex (1 = male; 2 = female), country of residence (Croatia, Greece, Spain), educational achievement (primary, secondary, higher), locality of residence (urban, rural). Second, the motivation factors: health, environment & animal welfare as binary variables (1 = yes, 0 = no). Third, the perception about cultured meat (healthiness, environmental impact, taste, animal welfare, naturalness, and preconceptions) all on a 5-point Likert Scale from strongly disagree to strongly agree. Fourth, the intention of trying or tasting on a 5-point Likert Scale from very unlikely to very likely, and the intention of purchasing at the same price, higher price, or lower price as binary variables (1 = yes, 0 = no). Binary logistic regressions in R uses the maximum likelihood estimation method, in this case to estimate the log odds of the dependent variable taking the value of 1, meaning belonging to the cluster. Results are presented as odds ratios (ORs), which express the likelihood of belonging to each of the clusters by each variable keeping the others constant. Models for motivation, awareness, perception, and intention were adjusted for age, sex, country, education, and type of residence. The open question was analyzed in R using the text mining technique. 

### 2.3. Data Management 

Data management was performed following a standard procedure. SurveyXact creates a dummy variable 1 = complete; 0 = incomplete. Only answers that were complete were further retained. Thereafter, consistency was checked, e.g., if respondents declared being vegans and had some frequency of meat intake, then these would be removed. Respondents under the age of 18 were removed, and only respondents in the three countries were retained. 

## 3. Results

A total of 2638 questionnaires were collected. Based on the inclusion criteria, 631 answers were excluded from further processing. Therefore, 2007 answers were analyzed. 1444 participants were females (72%) and 563 were males (28%). The age of the participants ranged from 18 to 88 years. Eighty-seven percent of the respondents lived in urban/suburban areas, and 13% in rural areas. Sixty-seven percent of the participants had completed university education, 26% high school education, 6% vocational training, and 1% elementary education. Forty-five percent of the respondents were Greek (*n* = 901), 28% were Croatian (*n* = 565), and 27% Spanish (*n* = 541).

Three segments of meat consumers were created, plus an a priori defined group of “non-meat eaters”. The “non-meat eaters” group constituted 3.6% (*n* = 72) of the total sample. Meat eaters represented 96.4% (*n* = 1935) of the sample. Meat consumption frequency ranged from 0 to 5.57 times per day. This means that consumers eat meat on average 6 times per week. The first meat consumers’ segment is characterized by a medium frequency of meat consumption (0.53 times/day) and low processed meat consumption (0.12 times/day). This segment accounts for the 50.2% of the sample and will be subsequently referred to as the “medium frequency, low processed” segment. The second segment of meat consumers is defined by high meat consumption (1.3 per day, which means more than once a day) and medium processed meat consumption (0.4 times/day). This segment will be subsequently referred to as the “high frequency, medium processed” segment, which accounts for 41.8% of the sample. The last segment, which accounts for 4.4% of the sample, consists of meat consumers with the highest frequency of meat consumption (2.9 times/day) and a high processed meat consumption (0.93 times/day, almost once per day). This segment will be referred to as the “very high frequency, high processed” segment.

### 3.1. Sociodemographic Characteristics

Table 1 shows the distribution of the studied population. It presents personal differences between non-meat eaters and each of the three meat eater segments based on analysis using a chi-square test. Segment membership showed significant association with sex (0.002), age (<0.001), and nationality (<0.001) while residence or education did not show any association.

Table 2 presents the sociodemographic characteristics across the different clusters. Regarding the “non-meat eaters” segment, it is 2.8 times more likely for a female to belong to this segment than for a male. Croatians are 75% less likely to belong to this segment, followed by Greeks (44% less likely), when compared to Spain. When describing the “medium frequency, low processed” segment, it is 27% more probable for women to belong to this segment compared to men. There are statistically significantly fewer Croatians in this segment, compared to Spaniards and Greeks. The majority of the participants who had finished elementary school were found in this cluster (12 out of a total of 16). For the “high frequency, medium processed” segment, it is 21% less likely for women to belong to this group. Country-wise, people from Croatia are 49% more likely to belong in this segment. The cluster “very high frequency, high processed” seems to be very heterogeneous since it is not described by any of the demographic variables.

As far as age is concerned, each increase in age by 1 year, leads to a 2% decrease in the likelihood of belonging to the “high frequency, medium processed” and a 4% decrease in the “very high frequency, high processed” segment, while it is a 2% increase for the “medium frequency, low processed meat” segment. For the “non-meat eaters”, the likelihood decreases by 4% for every additional year of age, indicating that younger people are more likely a part of this group. The education level and residence area do not seem to describe any of the segments as the results are not statistically significant. For the total sample of participants, percentages of motivations to eat less meat are 76% for health reasons, 40% for animal welfare, and 32% for environmental reasons. 

### 3.2. Attitudes and Intentions towards Cultured Meat

Table 3 presents the motivating factors that could lead (or have already led) participants to consume less meat. Animal welfare and the environment are more likely to be the reasons that “non-meat eaters” have eliminated their meat consumption, with odds ratios of 8.00 and 6.11, respectively. They are 63% less likely to have done it for health reasons. In the “medium frequency, low processed” segment, environmental reasons are twice as likely to be the motivation, while animal welfare comes second with a percentage of 42%. It is 29% less likely for the participants to lower their meat consumption due to health reasons. On the contrary, health is 36% more likely to motivate respondents belonging to the “high frequency, medium processed” group. Animal welfare is 73% less likely to be the motivating reason here. In the “very high frequency, high processed” segment, environmental reasons are 59% less likely to motivate the respondents. Overall, animal welfare and environment seem to be more important for the “non-meat eaters” and the “medium frequency, low processed” segments, while health comes first for the other two meat-consuming groups.

Table 4 presents participants’ awareness about the term “cultured meat”. In total, 47% of the participants had not heard of the term before, 41% had heard about it, and 12% knew what it was. Results demonstrate that “non-meat eaters” are most aware of the term.

Table 5 displays the participant’s perception of cultured meat. The perception that cultured meat could be healthy and tasty is 21% more likely to come from the “non-meat eaters”, while its kindness to animals is 10% more likely to appear as an answer among respondents of the “high frequency, medium processed” segment. Of the total number of participants, 60% thought that cultured meat is kind to animals, 57% that it is unnatural, 45% that it is healthy and environmentally friendly, 21% that it is disgusting, and only 16% that it is tasty.

Table 6 demonstrates participants’ intentions of trying (tasting) and purchasing cultured meat. Tasting is 29% less likely among the “non-meat eaters” but has a 9% positive association with the “medium frequency, low processed” segment. The “high frequency, medium processed” segment would buy cultured meat at any price, while the other segments are less likely to buy it. “Non-meat eaters” have a 69% negative correlation with purchasing at a higher price. Overall, 43.5% of the participants would taste cultured meat, 53% would purchase it at a lower price compared to conventional meat, 41% at the same price, and 17% at the higher price.

The 10 most frequent words in the open question were: artificial (284 times), laboratory (173 times), unnatural (112 times), processed (96 times), disgusting (62 times), GMO (62 times), unknown (55 times), chemicals (54 times), unhealthy (51 times), nothing (49 times). Other words that appeared multiple times were animal welfare, fake, food, weird, cells, interesting, not tasty, and kind to animals. 

## 4. Discussion

### 4.1. Awareness and Perceptions towards Cultured Meat

Almost half of the participants (47%) had never heard of the term “cultured meat” before, which demonstrates low rates of awareness among the general population of these 3 countries. Only 12% responded with “I know what cultured meat is” and these were most likely to be non-meat eaters. The majority of the respondents (60%) thought of cultured meat as being kind to animals. “Healthy” and “environmentally friendly” scored a percentage of 45. Sixteen percent of the participants described cultured meat as being “tasty” demonstrating that appeal of cultured meat is generally perceived as low. Results on perception did not significantly differ for the segments, except for higher possibilities for “healthy” and “tasty” responses to belong to the “non-meat eaters”. The “non-meat eaters were also 29% less likely to try cultured meat and 69% less likely to purchase it at a higher price. This is in accordance with the research of Wilks et al. [11], in which they found that vegetarians and vegans had more positive perceptions of some aspects of cultured meat but that they were significantly less willing to consume it than omnivores. 

The word count tends to highlight the “artificial”, “laboratory-made”, and “unnatural” perception for cultured meat, adding up to an ambiguous “fear of the unknown” aspect, as supported by Verbeke et al. [13]. Only 21% of the study participants thought that cultured meat was disgusting, but 57% of them described it as “unnatural”. In addition, it was obvious from some replies to the open question, that some participants did not understand the concept of cultured meat. This could be because only a brief definition of cultured meat was given and no other wording for cultured meat was used. Furthermore, in this study, the overall acceptance could be characterized as high if intentions to taste and purchase are taken into consideration. This is in accordance with the findings of Bryant and Barnett [14] who state that the less information potential customers have about the technology of production, the less disgust they feel. Further research is needed to confirm whether this theory is correct. Furthermore, “less negative” wording like ‘clean meat’, ‘pure meat’, or ‘meat without animal suffering’ could lower the disgust and the feeling of unnaturalness. On the contrary, the negatively-associated wording like synthetic meat, in vitro meat and artificial meat could further increase the feeling of disgust and unnaturalness.

### 4.2. Intentions towards Cultured Meat

In this study, 43.5% of the participants indicated that they would try cultured meat. A study conducted by Bryant et al. [15] showed similar replies regarding trying cultured meat (43.5% of the total respondents). Similar results were demonstrated by Weinrich et al. [16], where 57% of Germans intended to try it and 30% would eat it on a regular basis, based on a 5-point scale. Participants from Italy and the USA seem to have similar opinions on trying cultured meat, 54% and 65.3%, respectively [12,17].

Forty-one percent of the participants from this research would purchase cultured meat for the same price as conventional meat and 53% would buy it for a lower price. These results indicate that half of the potential consumers would buy a product made of cultured meat, but that the price is a crucial factor. However, it was interesting to notice that the “high consumption, medium processed” group would purchase cultured meat at any price. “Non-meat eaters” were 29% less likely to taste cultured meat and 69% less likely to purchasing at a higher price. That information describes the potential target group of the cultured meat industry.

Although the acceptance of cultured meat according to this research study seems promising for the European market, more steps should be taken towards overcoming the existing concerns and doubts. Developers of cultured meat support the fact that once the manufacturing process has been developed completely and the technology is explained to the people, consumer rejection may further decrease. In addition, if the potential consumers were more aware of animal slaughtering and the negative environmental impacts of the meat industry, they would reduce or cease their meat consumption [17,18]. Finally, if sensory qualities more similar to conventional meat were achieved, acceptance would further rise [19,20].

### 4.3. Public Health Nutrition Implications

From a public health nutrition perspective, there is consensus on the fact that consumption of foods of animal origin should be reduced dramatically in most countries for environmental [3,9,21,22,23,24] and public health reasons [6,7,8,25,26,27,28]. Given that this transition requires innovative solutions, a way ahead could be that large segments of the population would consume cultured meat, in small quantities as recommended, e.g., by EAT-Lancet [21] and providing the consumer with the same hedonic experience [29].

Low or non-meat eaters were more likely to be women in this sample of respondents. This agrees with previous studies where women reported low or no meat consumption and greener behavior [30]. Moreover, Croatian respondents were less likely to belong to the low or non-meat eater groups, as it seems that environmental issues are not associated to food choices in Croatia [31]. The group of heavy meat eaters was not explained by any sociodemographic characteristic. This suggests that this behavior is widely spread in the studied countries, which is supported by evidence indicating that adherence to the plant-rich Mediterranean diet is low [32] and that consumption of fruits and vegetables is below recommendations in EU countries [33].

Sustainable and healthy eating can be achieved if populations move towards a more plant-rich diet [22]. The future promotion of more sustainable food consumption may include products such as cultured meat in the portfolio [1]. According to our findings the population segment that might be more likely to engage in behavior change is the one with high meat consumption and with moderate processing level. This will automatically reduce the overall red meat intake with the potential health benefits associated to this reduction [25,26,27], while still delivering the deliciousness of the umami taste that is appreciated by consumers [29].

### 4.4. Strengths and Limitations of the Study

The large sample size and the cultural diversity of this study are both considered strengths. This survey may also bring substantial information about the topic due to the novelty of the topic and the limited previously conducted research, particularly in Croatia and Greece. However, since the survey is based on a convenience sample, obtained through snow-ball procedure in different social media, it resulted in a sample of mainly females, university-educated, living in urban or suburban areas. One way of controlling for sampling error is to apply regression methods, as they evaluate the influence of variables *‘ceteris paribus’* or keeping the others constant. Therefore, the results account for the sampling bias in the modelling. Nevertheless, it is a limitation of the study, and results need to be understood in the context of the current sample. Over-representation of women can be desirable in studies where new foods are introduced as, in general, they are the main actor in meal and diet lifestyle choices for families in traditional societies such as in Croatia, Greece, and Spain. For mainstreaming the consumption of innovative products, it is important to understand where women stand towards such innovations, hence, the lessons of this cross-sectional study will inform both nutrition policy and product development. When evaluating external limitations, respondents may have under- or over-reported their meat consumption. Lastly, although a definition of cultured meat was given, it was obvious from some replies to the open question that some respondents did not understand the concept. This could have influenced their responses regarding their opinions and intentions.

## 5. Conclusions

The key insights of the study are: First, meat-avoiding or low frequency meat consumers are mostly motivated by the environment and animal welfare, while the other segments are mostly motivated by health in their pursuit of meat reduction. Second, the implication of the study is that awareness about cultured meat in this sample of respondents is still relatively low (47%) and mainly concentrated among those in the lowest meat consumption segments. Third, non-meat eaters are less likely to try or purchase this particular product. The study suggests that the segment with high frequency of moderately processed meat consumption might be more prone to shifting towards cultured meat alternatives in the future. 

The directions for future research in the field that are relevant for public health nutrition include the evaluation of long-term healthiness of such products, the adequate intake levels, and how to improve awareness at population level to facilitate the shift. 

This study suggests that consumers in Croatia, Greece, and Spain are likely to purchase cultured meat if it is affordable. Moreover, efforts should be made to increase mainstream consumer awareness about the environmental, ethical, and health aspects related to cultured meat to prepare citizens and the market for this innovation.

## Figures and Tables

**Table 1 nutrients-13-01284-t001:** Sociodemographic characteristics.

	Non-Meat Eaters	Medium Frequency, Low Processed	High Frequency, Medium Processed	Very High Frequency, High Processed	
	*n* = 72 (3.6%)	*n* = 1008 (50.2%)	*n* = 839 (41.8%)	*n* = 88 (4.4%)	*p* Value
Age mean (SD)	29(11)	35(13)	32(11)	30(10)	<0.001 ***
Sex Female (%)	63(87.3)	743(73.7)	577(68.8)	61(69.3)	0.002 **
Country *n*(%)					<0.001 ***
Spain	32(44)	271(26.9)	220(25.0)	28(31.8)	
Croatia	9(12.5)	241(23.9)	285(34.0)	30(34.1)	
Greece	31(43.1)	496(49.2)	344(41.0)	30(34.1)	
Residence Rural *n*(%)	9(12.5)	126(12.5)	110(13.1)	12(13.6)	0.975
Education *n*(%)					
Primary	1(1.4)	12(1.2)	3(0.4)	0(0.0)	
Secondary	14(19.4)	235(25.1)	222(26.5)	25(28.4)	
Vocational training	3(4.2)	59(5.9)	51(6.1)	11(12.5)	
University	54(75.0)	684(67.9)	563(67.1)	52(59.1)	0.103

Asterisk indicates statistical significance: ** <0.01, *** <0.001.

**Table 2 nutrients-13-01284-t002:** Sociodemographic characteristics; logistic regression results.

		Non-Meat Eaters		Medium Frequency, Low Processed		High Frequency, Medium Processed		Very High Frequency, High Processed	
		*n* = 72 (3.6%)		*n* = 1008 (50.2%)		*n* = 839 (41.8%)		*n* = 88 (4.4%)	
		OR (95% CI)	*p* Value	OR (95% CI)	*p* Value	OR (95% CI)	*p*-Value	OR (95% CI)	*p* Value
Age	1 year increment	0.96 (0.94–0.99)	0.010 *	1.02 (1.01–1.03)	<0.001 ***	0.98 (0.97–0.98)	<0.001 ***	0.96 (0.94–0.98)	<0.001 ***
**Sex**	Male [REF]	1		1		1		1	
	Female	2.80 (1.46–6.1)	<0.001 ***	1.27 (1.04–1.54)	0.017 *	0.79 (0.66–0.97)	0.027 *	0.90 (0.57–1.45)	0.663
**Country**									
	Spain [REF]	1		1		1		1	
	Croatia	0.25 (0.11–0.52)	<0.001 ***	0.67 (0.52–0.85)	0.001 **	1.49 (1.17–1.90)	0.001 **	0.97 (0.57–1.67)	0.939
	Greece	0.56 (0.34–0.94)	0.027 *	1.16 (0.93–1.45)	0.174	0.93 (0.74–1.16)	0.53	0.61 (0.36–1.04)	0.069
Residence									
	Urban/suburban [REF]	1		1		1		1	
	Rural	0.97 (0.44–1.87)	0.937	0.94 (0.72–1.23)	0.664	1.04 (0.79–1.36)	0.735	1.07 (0.55–1.93)	0.814
Education									
	Primary	2.38 (0.12–13.10)	0.417	3.90 (1.22–17.29)	0.036 *	0.31 (0.07–0.99)	0.074	1.21 (NA)	0.982
	Secondary [REF]	1		1		1		1	
	Vocational training	0.88 (0.20–2.76)	0,85	0.92 (0.62–1.38)	0.716	0.91 (0.60–1.36)	0.654	1.90 (0.87–3.88)	0.088
	University	1.48 (0.84–2.80)	0.194	1.08 (0.88–1.33)	0.434	0.95 (0.77–1.17)	0.685	0.79 (0.87–3.88)	0.35

Asterisk indicates statistical significance: * <0.05, ** <0.01, *** <0.001.

**Table 3 nutrients-13-01284-t003:** Motivational factors that may lead participants to consume less meat.

	Non-Meat Eaters		Medium Frequency, Low Processed		High Frequency, Medium Processed		Very High Frequency, High Processed	
	OR (95% CI)	*p* Value	OR (95% CI)	*p* Value	OR (95% CI)	*p* Value	OR (95% CI)	*p* Value
Animal welfare	8.00 (4.31–16.23)	<0.001 ***	1.42 (1.17–1.72)	<0.001 ***	0.73 (0.60–0.88)	0.001 **	0.82 (0.52–1.29)	0.40
Environmental reasons	6.11 (3.55–10.99)	<0.001 ***	2.00 (1.63–2.47)	<0.001 ***	0.54 (0.44–0.67)	<0.001 ***	0.59 (0.35–0.97)	0.043 *
Health reasons	0.37 (0.23–0.59)	<0.001 ***	0.71 (0.57–0.88)	0.002 **	1.36 (1.09–1.69)	0.007 **	1.29 (0.77–2.24)	0.35

An odds ratio (OR) > 1 indicates that respondents are more likely to consume less meat because of the given reasons, while an OR < 1 indicates that consumers in the cluster are less likely to consume less meat because of the given reasons. Results are adjusted for age, sex, country, education, and residence. Asterisk indicates statistical significance: * <0.05, ** <0.01, *** <0.001.

**Table 4 nutrients-13-01284-t004:** Participants’ awareness about the term cultured meat.

	Non-Meat Eaters		Medium Frequency, Low Processed		High Frequency, Medium Processed		Very High Frequency, High Processed	
	OR (95% CI)	*p* value	OR (95% CI)	*p* Value	OR (95% CI)	*p* Value	OR (95% CI)	*p* Value
Awareness	1.54 (1.23–1.91)	<0.001 ***	1.11 (1.00–1.22)	0.048 *	0.89 (0.81–0.98)	0.023 *	1.07 (0.85–1.33)	0.542

An OR > 1 indicates that the consumers in the cluster are familiar with the term cultured meat, while an OR < 1 indicates that consumers in the cluster are less aware of the term. Results are adjusted for age, sex, country, education, and residence. Asterisk indicates statistical significance: * <0.05, *** <0.001.

**Table 5 nutrients-13-01284-t005:** Participants’ perceptions about cultured meat.

	Non-Meat Eaters		Medium Frequency, Low Processed		High Frequency, Medium Processed		Very High Frequency, High Processed	
	OR (95% CI)	*p* Value	OR (95% CI)	*p*-Value	OR (95% CI)	*p* Value	OR (95% CI)	*p* Value
Healthy	1.21 (1.00–1.67)	0.049 *	0.99 (0.90–1.09)	0.922	0.99 (0.89–1.09)	0.787	1.13 (0.88–1.04)	0.36
Environmentally friendly	1.23 (0.99–1.61)	0.071	1.00 (0.92–1.09)	0.933	1.00 (0.92–1.09)	0.995	0.98 (0.79–1.21)	0.831
Tasty	1.60 (1.22–2.11)	0.001 **	1.08 (0.99–1.20)	0.09	0.93 (0.85–1.03)	0.17	0.92 (0.73–1.15)	0.451
Kind to animals	0.86 (0.69–1.07)	0.173	0.92 (0.84–1.01)	0.067	1.09 (1.01–1.20)	0.039 *	0.95 (0.78–1.17)	0.61
Unnatural	0.88 (0.71–1.09)	0.247	0.95 (0.87–1.03)	0.239	1.09 (1.00–1.18)	0.057	0.85 (0.69–1.02)	0.08
Disgusting	0.80 (0.61–1.02)	0.079	0.93 (0.85–1.02)	0.134	1.07 (0.97–1.18)	0.151	1.01 (0.81–1.27)	0.867

An OR > 1 indicates that the consumers in the cluster agree with the given statement, while an OR < 1 indicates that consumers in the cluster disagree. Results are adjusted for age, sex, country, education, and residence. Asterisk indicates statistical significance: * <0.05, ** <0.01.

**Table 6 nutrients-13-01284-t006:** Participants’ intentions of trying cultured meat.

	Non-Meat Eaters		Medium Frequency, Low Processed		High Frequency, Medium Processed		Very High Frequency, High Processed	
	OR (95% CI)	*p* Value	OR (95% CI)	*p* Value	OR (95% CI)	*p* Value	OR (95% CI)	*p* Value
Tasting	0.71 (0.57–0.87)	0.002 **	1.09 (1.00–1.18)	0.043 *	0.93 (0.85–1.01)	0.075	0.94 (0.78–1.14)	0.542
Purchase								
Same price	0.78 (0.48–1.28)	0.327	0.74 (0.61–0.90)	0.003 **	1.36 (1.12–1.65)	0.002**	0.95 (0.61–1.49)	0.828
Higher price	0.31 (0.19–0.52)	<0.001 ***	0.71 (0.55–0.91)	0.008 **	1.34 (1.04–1.73)	0.021*	1.28 (0.72–2.49)	0.408
Lower price	1.10 (0.67–1.81)	0.697	0.90 (0.74–1.09)	0.263	1.14 (0.94–1.38)	0.002**	0.91 (0.57–1.43)	0.688

An OR > 1 indicates that consumers are more likely to taste and purchase cultured meat, while an OR < 1 indicates that consumers in the cluster are less likely to try and buy cultured meat. Results are adjusted for age, sex, country, education, and residence. Asterisk indicates statistical significance: * <0.05, ** <0.01, *** <0.001.

## Data Availability

Following GDPR, data is stored in anonymous manner. Data is available upon request from the authors.

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
