# Peer review of "How Do Consumers Perceive Cultured Meat in Croatia, Greece, and Spain?"

_nutrients, 2021, doi:10.3390/nu13041284_

Round 1

Reviewer 1 Report

Dear Authors,

The manuscript (nutrients-1174828) presented for review is very interesting .

Authors, Please note and address the following comments:

Introduction

This is a very interesting topic which indicates a new meat approach in a few of chosen countries. Interest in meat is currently changing due to an increase in the number of vegetarians all the world, as well as ethical and health concerns connected with meat.

There is a lack of information if the cultured meat is possible to buy in Croatia, Greece, and Spain, or maybe is it only laboratory research connected with this meat? This information could have influenced on answers of consumers.

Where did the authors state that segmentation of consumers is an aim or objective of the study?

Material and Methods

In supplementary material should add a questionnaire, because difficult to understand which questions connected with meat were asked for consumers. Only many questions connected with the characteristics of consumers are well presented.

Results

Table 1/ Table 2 - I am not sure what is the number in the parentheses. Could authors explain these below Tables?

Table 2 – In row (line) “Residence” is only value 1 for non-meat eaters, what about other groups?

Figure 1 - Although Figure 1 looks attractive, in my opinion, should be omitted because it adds nothing, and the description in the text is sufficient.

Conclusion

What are the implications of the findings?

What should be the directions of further research in this range?

The current conclusions are quite enigmatic.

Despite my comments, I am pleased to recommend this manuscript for publication. I believe it addresses an important area of research in an international context.

Reviewer

Reviewer 2 Report

More information on the method section is required to proceed with the peer review.

Section 2.1

Include SurveyXact manufacturer and country of origin

Was there any ethics statement for this?

Consider creating a table for readers' clarity on what dimensions were collected and the anchors are.

Section 2.2

Include RStudio 1.3 manufacturer and country of origin

Table 5. How did these dimensions collected? Some info are missing.

How did the authors generally test their data validity and questionnaire reliability?

How did the authors calcuate OR?

Did the authors run some data cleanup? For example removing observations that did not made sense or was done too fast?

Was there a specific inclusion or exclusion criteria for the participants?

Round 2

Reviewer 2 Report

The authors have addressed the comments especially on the methodology part. However, this needs to be added back to the manuscript.

Discussion seems a bit sparse, I'd recommend the authors to create subsections and highlight their key results. What were the key insights and highlights of their study?
